# Dysregulation of MiR-21, MiR-221 and MiR-451 During Neoadjuvant Treatment of Breast Cancer: A Prospective Study

**DOI:** 10.3390/biom14121580

**Published:** 2024-12-11

**Authors:** Carine Bispo Patury, Brenda Luanny Maia Santos, Anna Lucia Carvalho Matos, José Slabi, Luciene Cristina Campos Gastalho, Carla Martins Kaneto

**Affiliations:** 1Department of Health Science, Universidade Estadual de Santa Cruz, Ilhéus 45662-900, BA, Brazil; carinepatury@gmail.com (C.B.P.); slaibios@gmail.com (J.S.); 2Department of Biological Science, Universidade Estadual de Santa Cruz, Ilhéus 45662-900, BA, Brazil; brendamaia228@gmail.com (B.L.M.S.); annaluciaacm@gmail.com (A.L.C.M.); lcgcluiz@uesc.br (L.C.C.G.)

**Keywords:** breast cancer, neoadjuvant treatment, molecular biomarker, microRNA

## Abstract

Background: Breast cancer is highly heterogeneous disease in which different responses are observed to the same treatment for different subtypes and extents of similar diseases. Considering this scenario, the search for tumor biomarkers is indispensable, with current evidence suggesting that circulating microRNAs are viable biomarkers. This study evaluated the expression of miR-21, miR-221, miR-195, and miR-451 in patients with breast cancer undergoing neoadjuvant treatment at oncology outpatient facilities in Brazil. Methods: We conducted a prospective and observational study in which blood samples were collected for microRNA expression analysis, comparing control and breast cancer patients who were candidates for neoadjuvant treatment groups. The expression of microRNAs was investigated by qRT-PCR method. For parametric data analysis, one-way ANOVA with Tukey’s post hoc test was used. Results: Thirty-three participants (all female) were included in the control group and twenty-seven participants were included in the study group. The non-special subtype of breast cancer was found in 96% of the study group participants; 88.9% were locally advanced tumors (T3, T4), 40.7% were luminal tumors, 33.3% were HER-2-positive, and 26% were triple negative tumors. Expression analysis of microRNAs during neoadjuvant treatment, using miR-16 as a normalizer, showed higher expression levels of miR-21 and miR-221 at the end of treatment, and high expression levels for miR-451 were also observed at the beginning of treatment. Conclusion: This is the first study that evaluates the expression of microRNAs in the context of neoadjuvant treatment of breast cancer in the Brazilian population. Our results suggest that there is a deregulation of miR-21, miR-221, and miR-451 during neoadjuvant treatment in these patients.

## 1. Introduction

Worldwide, and apart for non-melanoma skin cancer, breast cancer represents the highest incidence and the main cause of mortality in women [1]. In Brazil, according to the National Cancer Institute, estimates for 2023 are about 73.610,000 new cases of breast cancer [2]. When considering invasive breast cancer as a heterogeneous disease, its histological and molecular classifications are broad. The most recent edition of the World Health Organization for breast cancer subdivides breast carcinomas into 19 histological subtypes, of which 70% of the cases correspond to the non-special subtype, 10–15% of the cases correspond to the lobular subtype, and the others represented by rare subtypes [3]. Regarding the molecular categorization, as determined by the expression of hormone receptors (estrogen and progesterone), the product of the HER-2 oncogene, and the expression of ki67 in the immunohistochemical analysis, five subtypes are currently recognized. They are luminal A, luminal B, luminal B-HER-2-positive, enriched HER2 (non-luminal), and triple negative [3].

With the advancement of systemic treatment and incorporation of new therapies, along with the better clinical outcomes associated with complete pathological response (absence of residual tumor after oncological treatment), neoadjuvant treatment (performed prior to surgery) has been increasingly used [4]. In addition to enabling the increase in breast- and axillary-conserving surgeries, the neoadjuvant treatment of invasive breast cancer allows the evaluation of tumor behavior “in vivo”, the treatment of micrometastatic disease, time to search for germline mutations, and best surgical definition. It is noteworthy that response rates to neoadjuvant treatment are variable according to tumor phenotype, with complete pathological response rates ranging from 7–16% for tumors with positive hormone receptors (higher for high-grade tumors), 34% for triple-negative tumors, 30% for luminal tumors B-HER-2-positive (hormone receptor and HER-2 gene product expression-positive), and about 50% when tumors present expression of the HER-2 gene product and negative hormone receptors. Joint analyses of studies demonstrate that the complete pathological response after neoadjuvant treatment has evidence of improved survival free of oncological events related to breast cancer and overall survival, especially for triple negative tumors and isolated HER-2-positive [4,5,6].

However, due to the tumor heterogeneity of breast cancer, tumors of the same subtype and staging may reflect different responses to the same treatment. This fact, added to the unavailability of methods of high accuracy and low cost that can predict which patients will benefit from certain therapies, makes the search for tumor biomarkers indispensable. In this scenario, the study of the expression of circulating microRNAs in patients with breast cancer has advanced, with evidence suggesting that they are viable circulating biomarkers that are easily identifiable and that can be monitored in a minimally invasive way [7].

MicroRNAs, small non-coding RNAs of up to 22 nucleotides, were first described in 1993 in *Caenorhabditis elegans*, in which it was found that an RNA of small size could be associated with the time of development of this organism. Only in 2002, through the work of Drs. George Callin and Carlo Croce in Chronic Lymphocytic Leukemia (CLL), was the first evidence of the involvement of microRNAs in the pathogenesis of cancer noticed [8].

Due to the greater stability of the microRNA molecule compared to mRNA, its analysis in tissues and other biological materials (blood, urine) supports the idea of it being a viable model as a new biomarker [9,10,11]. The existence of circulating microRNAs was confirmed in 2008, with important differences between control patients and cancer patients.

In the scenario of breast cancer, circulating microRNAs have been used as a broad-spectrum biomarker, acting from diagnosis through to prognosis and as a predictor of response. Circulating microRNAs can still contribute to the development of targeted therapies, modifying the evolution of treatment and impacting on real gains for the patients involved [12,13]

In the context of neoadjuvant treatment of invasive breast cancer, an Irish study, published in 2020, evaluated the behavior of a panel of microRNAs prior to neoadjuvant treatment, in which the following were evidenced: reduced levels of miR-145, miR-21, and miR-195 in patients presenting a good response to treatment; miR-145 as a predictor of response in luminal subtype tumors and miR-21 as a predictor of response independent of tumor phenotype [14]. This confirms data from the Chinese study conducted in 2019 with patients with HER-2-positive tumors, in which reduced miR-21 expression was associated with the clinical response of the tumor but not the pathological response after neoadjuvant treatment [15].

Another study conducted in the Middle East, analyzing tissue and serum samples, showed that miR-451 presented higher expression in serum than in tissue, with a reduced gradual expression during neoadjuvant treatment, and is also implicated as a biomarker of clinical and pathological response, in addition to disease-free survival [16]. Also in this scenario, a study by Wenjie Zhu et al., published in 2018, demonstrated that reduced expression of miR-34a during treatment in both patients with HER-2-positive tumors and triple negative tumors was associated with no response to therapy. On the other hand, good responder patients evolved with stability in their expression compared to the evaluation performed prior to the neoadjuvant treatment [7]. The correlation between microRNA expression levels and complete pathological response in the HER-2-positive subtype with trastuzumab use was also evaluated by Serena Di Cosimo et al. in a study with 52 participants, in which the highest expression of miR-148a-3p and miR-374-5p was observed in good-responding participants [17]. MiR-221 expression was also shown to be associated with the expression of hormone receptors in tumors, in addition to its correlation with treatment resistance [18].

Despite some studies describing the relation between neoadjuvant treatment of breast cancer and dynamics of circulating microRNAs, the vast majority are studies conducted with a small number of patients carried out in the Eastern population, with chemotherapy regimens different from those performed in the Western world [7,19]. In addition, it is worth mentioning that there is no standardization in the evaluation of this biomarker in this scenario, with studies using diverse endogenous controls, samples, and collection moments. In this context, the aim of this study was to prospectively evaluate the behavior of the expression of circulating microRNAs (miR-21, miR-221, miR-195, miR-451) before, during, and after neoadjuvant treatment of invasive breast cancer.

## 2. Materials and Methods

### 2.1. Study Design

This is a prospective and observational study, with analysis of a study group (participants diagnosed with invasive breast cancer submitted to a neoadjuvant treatment program) and a control group (participants without previous oncological diagnosis), carried out in patients admitted in the Oncology Units of São José Hospital-Ilhéus-Bahia, the Women’s Health Center of Santa Casa de Jequié-Bahia, and the Laboratory of Applied Pathology and Genetics (LAPAGEN) of the State University of Santa Cruz-Ilhéus-Bahia.

### 2.2. Patients and Eligibility Criteria

All participants admitted to partner health services who met the inclusion criteria of the study were included in the study group: participants with breast cancer with diagnosis defined by biopsy; age over 18 years; with normal kidney, hepatic, cardiac (in those candidates for the use of anthracyclines, specifically), and medullary functions; with non-metastatic disease observed at the imaging exams (evaluation by total abdomen US and chest X-ray or chest and abdomen tomography, as well as bone scintigraphy) and without previous treatments for current disease (radiotherapy, hormone therapy, or chemotherapy) who were submitted to neoadjuvant treatment and who agreed to participate in the research, thus signing the written informed consent. All participants with another concomitant diagnosis of malignant neoplasm and patients with metastatic breast cancer were excluded.

The group of participants without previous oncological diagnosis (control group) was composed of volunteers recruited from the general population (most of the time, companions of outpatients of the selected services), without previous oncological diagnosis and who wished to participate in the research, signing the written informed consent.

### 2.3. Blood Samples, RNA Isolation and cDNA Synthesis

Blood samples (5 mL) were collected from each donor in BD vacutainers containing 10 mg dipotassium EDTA anticoagulant and processed by separation of the plasma, which was accomplished by centrifugation at 800× *g* for 10 min at room temperature, allowing cell debris removal. Supernatant plasma was recovered and those samples considered hemolyzed were excluded. A plasma sample of 300 µL was mixed with TrizolLS (Thermo Scientific, Waltham, Massachusetts, USA) with a ratio of 1:3 in a 1.5 mL microcentrifuge tube and incubated at room temperature for 5 min. Then, 250 µL of chloroform was added and mixed vigorously by vortexing. The mixture was incubated at room temperature for 15 min and centrifuged at 12,000 rpm for 20 min at 4 °C. The supernatant was transferred to a new microcentrifuge tube and RNA was precipitated by the addition of 800 µL isopropanol. After being incubated at −80 °C for 12 h and centrifuged at 12,000 rpm for 20 min at 4 °C, the RNA pellet was rinsed, air-dried, and resuspended in 20 µL RNAase-free water. RNA concentrations were determined with a NanoDrop 1000 (Thermo Scientific, Waltham, Massachusetts). Only RNA samples with a 260/280 ratio of ≥1.8 were included. Total RNA (500 ƞg) was reverse transcribed using miR-specific primers and Taqman miRNA Reverse Transcription Kit (Thermo Scientific, Waltham, Massachusetts) in a scaled down volume of 15 µL RT reaction, according to the manufacturer’s instructions. The thermal cycling parameters of reverse transcription were 30 min at 16 °C, 30 min at 42 °C, and 5 min at 85 °C. The cDNA samples were diluted in nuclease-free water and stored at −20 °C.

### 2.4. Quantitative Real-Time PCR

Expression levels of individual miRNAs were detected by subsequent RT-qPCR using Taqman MicroRNA assays (Applied Biosystems) and a QuantStudio3 Instrument (ThermoFisher Scientific) using standard thermal cycling conditions in accordance with manufacturer recommendations. RT-qPCR amplification mixtures contained 20 ƞg template cDNA, 10 µL Taqman master mix (Applied Biosystems), and probes for MiR-320a (assay ID: 002277), MiR-16 (assay ID: 000391), U6 (assay ID: 002571), miR-21 (assay ID: 000397), miR-221 (assay ID: 000494), miR-195 (assay ID: 000494), and miR-451 (assay ID: 001141) in a final volume of 20 µL. The PCR protocol was applied as follows: incubation for 10 min at 95 °C, followed by 40 cycles of 10 s at 95 °C and 1 min at 60 °C. The Ct values for RT-qPCR were determined using the QuantStudio™ Design & Analysis Software version 1.5.2 (Applied Biosystems) and the single-threshold method with normalization to an internal control miR-16 (Assay ID: 000391), using the 2 –ΔCt formula [20,21,22] and presented as fold change.

### 2.5. Selection of Candidate Gene and Stability Analysis

The expression of 3 miRNAs (miR-320a, miR-16, and U6) was detected in plasma samples of all subjects of each group. Their stability was analyzed by Reffinder (http://leonxie.esy.es/RefFinder/) (accessed on 21 October 2024) and NormFinder (https://moma.dk/normfinder-software) (accessed on 21 October 2024) software.

### 2.6. Ethical Aspects

This research was approved by the Research Ethics Committee (CEP) of State University of Santa Cruz, on registration in the Certificate of Presentation of Ethical Appreciation (CAAE) number 37084920.4.0000.5526.

### 2.7. Statistical Analysis

Parametric data were analyzed using one-way ANOVA with Tukey’s post hoc test. All data were analyzed using the Prism 5.01 software (GraphPad, San Diego, CA, USA). Statistical differences were considered significant when *p* < 0.05.

## 3. Results

### 3.1. Patients

A total of 27 participants, aged between 34 and 73 years (mean: 50.11 years with standard deviation of 9.72), were included in the study group. All patients were female, and only one participant was diagnosed with a tumor of the invasive lobular histological subtype identified in the biopsy, the others being carriers of tumors of the non-special histological subtype of invasive breast cancer. Only four participants were diagnosed by screening tests, and the others made the diagnosis through self-examination (nodule palpable by the patient). In most cases, the indication for oncological treatment was due to locally advanced disease, with only two cases for other reasons (indication of conservative surgery and molecular subtype).

With regard to the presence of diseases other than invasive breast cancer in the study group, 18 participants had comorbidities other than the oncological diagnosis, with a 10 participants with two or more associated diseases. Of the comorbidities reported, obesity (11 participants) and systemic arterial hypertension (9 participants) stand out. Of the patients who were obese, only two participants were known to be in menopause.

Regarding the tumor phenotype, 85.2% were grade 2 tumors, 88.9% were locally advanced tumors (T3, T4), but 51.8% of the total participants did not present clinical axillary involvement ipsilateral to the tumor. Regarding the molecular profile, 40.7% were luminal tumors (excluding luminal tumors B HER-2-positive), 33.3% had HER-2 protein expression, and 26% were triple negative tumors. The most used chemotherapy regimen was doxorubicin, cyclophosphamide, and paclitaxel (41.7%), followed in a tie by three other regimens, namely docetaxel together with cyclophosphamide; doxorubicin, cyclophosphamide, paclitaxel, and trastuzumab; and carboplatin, docetaxel, and trastuzumab (Table 1).

Regarding the data about the response to neoadjuvant treatment (Table 2), 12 participants presented complete clinical response. However, among these, only 50% obtained absence of invasive residual tumor in the anatomopathological analysis of the surgical specimen (complete pathological response). Considering the patients who presented complete pathological responses to neoadjuvant treatment, 50% had tumors of the Luminal B HER-2-positive subtype and the others had tumors of the triple negative subtype.

### 3.2. Identification of Candidate Gene and Stability Analysis

A total of 91 samples were included for the selection of a microRNA that could be used as an endogenous normalizer among those previously selected miR-16, miR-320a, and U6. Of these samples, 33 were from the control group and 58 from the study group (participants diagnosed with breast cancer), of which 22 were samples from the first collection (performed before neoadjuvant treatment), 21 from the second collection (samples collected after about 50% of the proposed neoadjuvant treatment had been performed), and 15 from the third collection (collected after the end of neoadjuvant treatment and before breast surgery).

Based on the data generated by the qRT-PCR experiment (Figure 1), the miR-16 presented the highest p-value and, consequently, the lowest statistical difference between the groups, being considered the most stable.

To verify the initial results, the same data obtained by qRT-PCR were submitted to an analysis with two tools that allow to obtain a more accurate result. The results obtained by RefFinder analysis were also favorable to miR-16 in most of the analysis parameters, as shown in Figure 2.

In the evaluation of the data generated by the Delta CT tool, the stability value of miR-16 was 3.93, lower than the values of miR-320a (4.19) and U6 (4.70), with miR-16 thus being considered the most stable microRNA by this evaluation. Regarding the data obtained by Genorm, U6 presented the highest stability value (4.28) when compared to miR-16 and miR-320a, which presented similar stability values equivalent to 3.42.

NormFinder analysis once again showed that miR-16 was the microRNA with the lowest stability value, equivalent to 1.85, with referent values for miR-320a and U6 of 2.87 and 4.04, respectively. When comparing the four tools used by RefFinder, the Bestkeeper software (version 1) was the only one that presented a stability value of U6 lower than the other candidate microRNAs, with the stability value of miR-16 being 3.90, that of miR-320a being 3.95, and that of U6 being 2.41. The sum of the mean values obtained in all these evaluations resulted in a global analysis performed by RefFinder, in which it can be inferred that miR-16 presented the lowest stability value, equivalent to 1.18, compared to miR-320a and U6, which presented stability values of 1.86 and 2.28, respectively. The isolated analysis by the NormFinder platform only corroborated the previous data, with miR-16 being considered the most appropriate endogenous control, with a stability value of 0.330 (Figure 3). Thus, based on all performed analysis, miR-16 was shown to be the best candidate microRNA as an endogenous control for the normalization of the samples involved in this study.

### 3.3. Evaluation of the Expression of MicroRNAs Selected in the Study Group During Neoadjuvant Treatment and in the Control Group

A total of 85 samples were included to evaluate the expression of the selected microRNAs, 29 samples belonging to the control group and 56 samples to the study group. Of the latter, 22 samples were from the first collection, 20 samples from the second collection, and 14 samples from the third collection. Due to the small number of samples, we decided to evaluate the expression of microRNAs selected by groups according to the moment of collection in relation to the treatment performed.

### 3.4. MicroRNAs Expression in the Study Group and Control Group

In the samples evaluated, and as represented in Figure 4, Figure 5 and Figure 6, there was a dysregulation of the expression of miR-21, miR-195, and miR-451 when comparing the control and study groups, with the latter two showing increased expression in the study group (both in the neoadjuvant pre-treatment collection-collection 1 -, with *p* < 0.05) compared to the control group. On the other hand, miR-21 expression was decreased in the study group when compared to the control group at the time of the first collection (before neoadjuvant treatment), with a *p* < 0.05. The difference between the groups remains statistically significant when comparing the study group with the control group in relation to the other moments of collection during the neoadjuvant treatment, presenting reduced expression when compared to the control group and second collection and increased when compared to the control group and third collection.

There was no statistical difference in the relative expression of miR-221 between the control group and the study group (Figure 7).

### 3.5. MicroRNAs Expression During Neoadjuvant Treatment of Breast Cancer

Regarding the behavior of microRNA expression during neoadjuvant treatment of breast cancer, there was an overexpression of miR-21 and miR-221 at the end of cancer treatment, with *p* < 0.005, as shown in Figure 4 and Figure 7.

Regarding the behavior of miR-451 during neoadjuvant treatment, there was an increased and significant expression (*p* < 0.05) of this microRNA before the start of treatment (collection 1) when compared to the other subgroups (collection 2 and 3). MiR-451 has a tendency to reduce its expression during cancer therapy, as shown in Figure 6.

There was no statistical difference in the relative expression of miR-195 during neoadjuvant treatment of invasive breast cancer, with only a tendency towards higher expression at the end of neoadjuvant treatment (Figure 5).

## 4. Discussion

Worldwide, breast cancer represents the highest incidence and the main cause of cancer mortality in women apart for malignant non-melanoma skin tumors [1]. Breast cancer is a heterogeneous disease, in which the decision about treatment in the non-metastatic setting is the result of a combination of factors, such as tumor phenotype, patient profile, and surgical proposal to be adopted. Given the data on survival gain in patients who present complete pathological response to treatment, neoadjuvant therapy is currently widely used, and its indication is no longer restricted to the treatment of locally advanced lesions in the breast [4,23,24]. However, the response to this treatment is still unpredictable, being higher among the triple negative and HER-2-positive subtypes [25]. As a result, the search for a biomarker that predicts response to neoadjuvant therapy is still necessary. MicroRNAs, small non-coding molecules of up to 22 nucleoids, responsible for acting in diverse cellular processes and in the regulation of gene expression, can be an interesting proposal as a minimally invasive biomarker in several pathologies, not being different in the oncology scenario. Several studies have demonstrated the role of circulating microRNAs as tools for diagnosis, response to treatment, and association with metastasis [26,27]. Although the current method of choice for quantifying the expression of circulating microRNAs is RT-PCR, there is still a risk of associated errors from sample collection and RNA extraction affecting the efficiency of the enzymatic process. Thus, the use of strategies to amplify the accuracy of the quantification of the expression of circulating microRNAs is fundamental, and the use of an endogenous control for this purpose is part of this process [28]. Currently, the literature has not established an adequate standardization for the use of normalizing microRNA, making it challenging to compare studies with the advancement of circulating microRNA as a biomarker.

Regarding the data found in our study, with regard to the diagnosis, a lower proportion of patients were diagnosed by screening tests (14.8%), corroborating data from the National Cancer Institute, in which breast cancer is diagnosed at an early clinical stage (tumor in situ and stage I) in about 20% of the cases [2]. Similar data were found in an Italian study, in which 28.3% of the patients were diagnosed with screening [29], as well as in the Brazilian study called AMAZONA, in which 34% of the total of 2950 patients evaluated made early detection of breast cancer. In the same study, the subgroup of patients who were part of the public health system obtained even higher rates of diagnosis for symptoms (palpable nodule), at 74.2% compared to 25.8% of the patients who were part of the private system [30].

About the tumor phenotype, 96.3% of the patients had breast carcinoma of a non-special subtype, which is usually the most frequent (about 70%) and with a more favorable response to neoadjuvant treatment with chemotherapy compared to the classic lobular tumor (luminal A profile). Unlike the data from the American study with patients with breast cancer undergoing neoadjuvant treatment by Boughey et al., in which 68% of the participants had tumors classified as T1 or T2, in our study, 88.9% had T3 or T4 tumors (Table 1) [25]. This may reflect the difficulty of access to screening tests and consequent diagnosis of advanced disease, as well as different rationales for choosing neoadjuvant treatment as opposed to post-surgery treatment. The non-availability of high-cost drugs in the adjuvant setting for some molecular subtypes that may increase the survival of patients with partial response to neoadjuvant therapy in the public health system in Brazil is an important factor that justifies the reason why most patients undergo neoadjuvant treatment when suffering locally advanced disease and not based only on the molecular subtype. It is noteworthy that despite the presence of locally advanced tumors in the breast, 51.8% of the participants in this study did not present ipsilateral axillary clinical involvement [31].

Considering the molecular profile, most participants (85.5%) had luminal tumor B, positive HER-2 (luminal B-HER-2 + or enriched HER-2), or triple negative, reflecting more aggressive tumors, with a higher rate of cell proliferation (81.5% had ki67 > 14%) and consequently a greater chance of responding to neoadjuvant treatment (Table 1).

Regarding the drugs used, as well as in the American study, most patients used a regimen containing anthracycline, 63% in our study (doxorubucin in all cases) and 74.9% in the study published by Boughey et al. [25]. This reflects the importance of this drug in the treatment of locally advanced breast cancer, as well as the presence of a smaller number of participants with HER-2-positive tumors (33.4% in our study), in which current evidence shows that anthracycline is not used in its respective treatment [32].

Regarding the response to the therapy described in Table 2, the complete pathological response rate (absence of residual invasive tumor in the breast and ipsilateral axillary) after neoadjuvant treatment was 22% in our study. These data are in agreement with the studies published by Boughey et al. and Mamtani et al., in which the rates of complete pathological response were 28.1% and 24%, respectively [33,34]. It is noteworthy that a small proportion of participants had a tumor with HER-2-positive subtype and negative hormone receptor in our study, and none presented a complete pathological response, despite the fact that these tumors exhibit the highest response rate described in the literature [3]. The unavailability of drugs that increase the complete pathological response in the neoadjuvant scenario in the public health system may have impacted this finding. In addition, the small number of samples may have hindered such an analysis.

Regarding the expression of circulating microRNAs among healthy patients and breast cancer patients, Wu et al., in 2012, demonstrated that miR-222 has increased expression in patients with breast cancer, with a sensitivity of 74% and specificity of 60% for the diagnosis of this pathology [35]. A Spanish study published in 2020 and a study conducted by Leidner et al. in 2013 associate a set of microRNAs that differentiate control samples from breast cancer samples, these being the association of the microRNAs miR-125b, miR-29c, miR-16, miR-1260, and miR-451 in the Spanish study and the microRNAs miR-708, miR-92b, and miR-568 in the study published by Leider et al. [36,37].

In this study, with regard to the expression data of the microRNAs previously selected (miR-21, miR-221, miR-195, and miR-451), for comparative analysis between the control group (without previous oncological diagnosis) and the study group (carriers of invasive breast cancer), it can be inferred that there was a statistical difference (*p* < 0.05) between the groups in terms of miR-21, miR-195, and miR-451. These last two microRNAs showed high expression in the study group compared to the control group. These data are corroborated by a Chinese study published in 2013, in which miR-451 presented elevated expression in the plasma of patients with breast cancer, as well as miR-16 and miR-21 [38]. However, data from the evaluation of miR-451 expression in the control groups and in patients with breast cancer who were candidates for neoadjuvant treatment demonstrated a reduced expression of this microRNA in the study group, in agreement with what was presented in our study [33,39].

With regard to miR-195, data from the literature are in agreement with those described in our study, with Zhao et al., in 2013, demonstrating a reduced expression of this microRNA in the serum of patients with breast cancer [40]. Cecener et al., in 2016, also demonstrated a reduction in miR-195 expression in breast cancer tissue samples compared with samples from normal controls. However, in the evaluation of plasma samples of this microRNA, there was a higher expression in patients with breast cancer before surgical treatment than after surgery [41].

Regarding miR-21, the data found in our study demonstrate a lower expression of this microRNA in the study group when compared to the control group (*p* < 0.005), different from the results found in the analyses of the microRNAs previously mentioned (Figure 4, Figure 5 and Figure 6). MiR-21 has been evaluated in several studies in breast cancer, in most of them presenting a high expression in breast cancer samples (blood or tissue) when compared to control samples from healthy individuals [42,43,44]. In this sense, we highlight the study conducted by Motawi et al. in 2016, in which the expression of miR-21 and miR-221 in the serum of women with breast cancer, benign breast lesions, and controls was evaluated, demonstrating an elevated expression of miR-21 in patients with invasive breast cancer [45]. A study conducted in Iran in 2018 confirms this data, evaluating tissue and plasma samples from 30 patients diagnosed with breast cancer and 10 controls and showing an equivalent expression between tissue and plasma of the microRNAs evaluated in the study group, with miR-21 among them, which was overexpressed in patients with breast cancer compared to the control group [46]. In parallel, Canatan et al. (2021) demonstrated that the expression of miR-21 can vary according to the selection of endogenous control, presenting overexpression in the breast cancer group when miR-181 was used as a normalizer and without statistical significance in its expression when miR-192 was used as an internal control [47]. Heneghan et al., in their study in which they used miR-16 as an endogenous control (similar to our study), also found no difference in miR-21 expression when comparing blood samples between controls and patients with breast cancer [48].

In contrast, a Chinese study published in 2019 by Chen et al., which evaluated serum samples from control patients with benign breast disease and breast cancer, showed a reduced expression of miR-21 in breast cancer patients, similar to the result found in our study [49].

Regarding the behavior of circulating microRNAs during neoadjuvant treatment of invasive breast cancer, the literature portrays a variety of microRNAs that are dysregulated in this context. However, there is no standardization in the evaluation of these small molecules, including in the type and time of sample collection, the endogenous control, and the type of chemotherapy regimens used. In our study, of the four microRNAs evaluated, only three showed statistical difference in their behavior during neoadjuvant therapy, specifically miR-21, miR-221, and miR-451. Regarding the latter, a study published by Zhu et al. in 2018 demonstrates that the expression of miR-451, miR-222, and miR-20a during neoadjuvant treatment of breast cancer is associated with sensitivity to chemotherapy in the luminal subtype HER-2-negative [7]. Data from a study conducted in Oman in 2016 corroborate the data found in our study, in which miR-451 presents a reduction in its expression during neoadjuvant treatment (Figure 6) [16].

Regarding the behavior of miR-221 during neoadjuvant treatment of breast cancer, in our study, this microRNA showed overexpression at the end of treatment (collection 3) compared to the beginning (collection 1), as shown in Figure 7. Gezer et al., in 2014, demonstrated a higher expression of miR-21, miR-195, and miR-221 in the plasma of participants with breast cancer before neoadjuvant treatment compared to other microRNAs included in the study, with miR-221 showing variation in its expression among participants during neoadjuvant treatment [50]. Zhao et al., in 2011, demonstrated that the overexpression of miR-221 was associated with the subtype of invasive breast cancer with negative hormone receptor and resistance to the therapy instituted [18].

Regarding the expression of miR-21 during neoadjuvant treatment of breast cancer, studies have demonstrated its potential use as a predictor of response to treatment [33,51], with its expression reduced at the end of therapy [15,52,53]. These findings differ from the results obtained in our study, in which miR-21 expression was higher at the end of neoadjuvant treatment (collection 3), as shown in Figure 4.

Differences found in the expression of the microRNAs selected in our study in relation to the data already reported in the literature may reflect issues associated with the use of different endogenous controls, the stage of disease of the patients included in the studies comparing control samples and breast cancer samples (some studies included patients with tumors in situ), and even issues related to ethnicity and race [54].

Finally, it is worth mentioning that our study has some limitations, such as a reduced number of samples evaluated, analysis of several subtypes of breast cancer at the same time, and the non-pairing of comorbidities between the control participants and the study group, the latter being able to directly impact the expression of microRNAs [55].

## 5. Conclusions

Our research represents a great contribution in the evaluation of the expression of microRNAs during the neoadjuvant treatment of breast cancer, being the first study in this profile to perform this evaluation in the Brazilian population, specifically in the Bahian population. Through it, we can suggest that there is a dysregulation of some of the microRNAs evaluated, both in the control samples when compared to the study group (miR-21, miR-195, and miR-451) and in the behavior of these small molecules during neoadjuvant treatment (miR-21, miR-221, and miR-451).

Studies that can validate data found in this research in the Brazilian population are necessary and should be carried out, with the possibility of generating a great scientific contribution in the area in question.

## Figures and Tables

**Figure 1 biomolecules-14-01580-f001:**
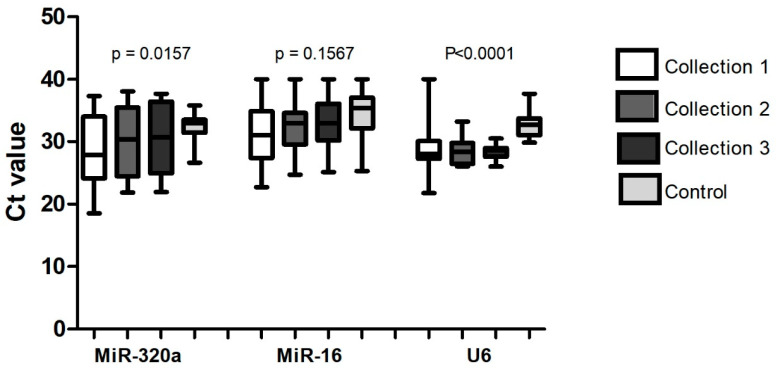
Analysis of the cycling threshold by qRT-PCR of miR-320a, miR-16, and U6 and their respective significance levels (*p*) in samples from the control group (n: 33) and the study group (n: 58). The evaluation was performed in 33 samples of the control group and in 58 samples of the group of patients with breast cancer, with collection 1 being performed before the neoadjuvant treatment (22 samples), collection 2 when 50% of the neoadjuvant treatment had been performed (21 samples), and collection 3 after the end of the neoadjuvant treatment (15 samples).

**Figure 2 biomolecules-14-01580-f002:**
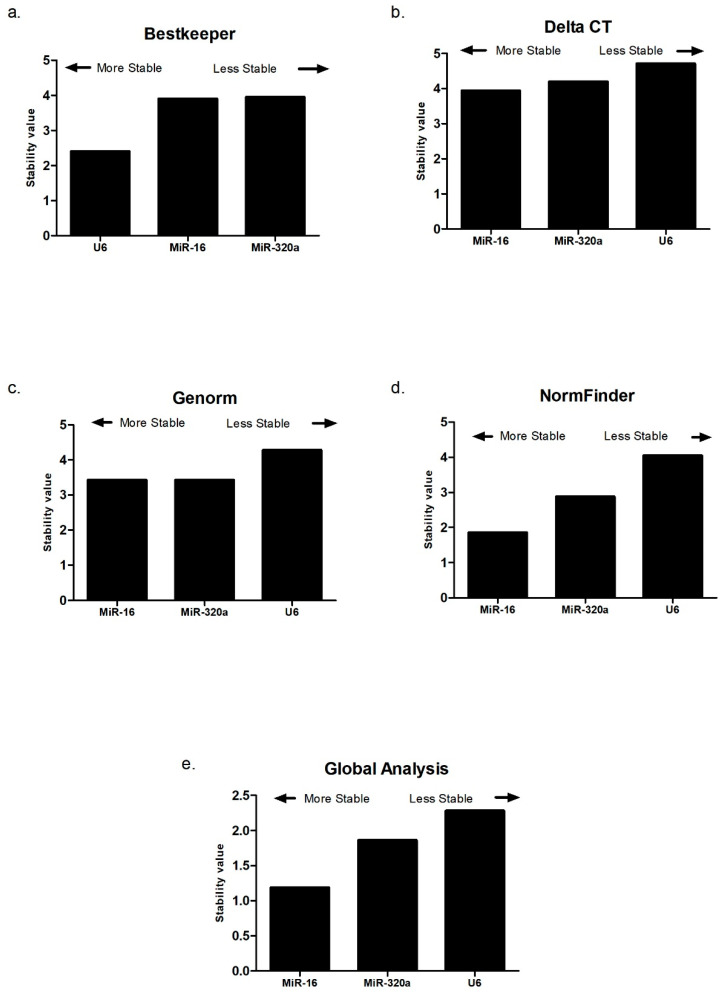
Stability levels of MiR-320a, MiR-16, and U6 evaluated in samples from the control group (n: 33) and the study group with breast cancer (n: 58) by several platforms and by Delta CT: (**a**) BestKeeper; (**b**) Delta CT (cycling threshold); (**c**) Genorm; (**d**) NormFinder; (**e**) Global analysis by RefFinder.

**Figure 3 biomolecules-14-01580-f003:**
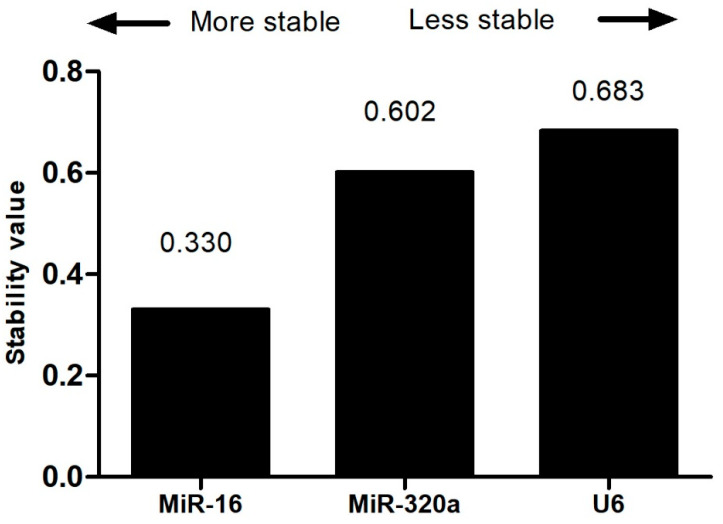
Stability levels of MiR-320a, MiR-16, and U6 evaluated in samples from the control group (n: 33) and the study group with breast cancer (n: 58) by the NormFinder platform.

**Figure 4 biomolecules-14-01580-f004:**
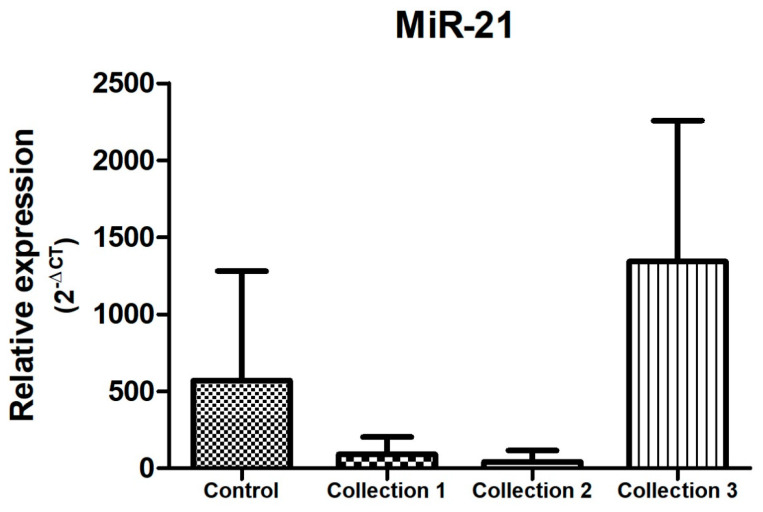
Relative expression of miR-21. Total N: 85 samples; control, n: 29; collection 1 (before neoadjuvant treatment), n: 22; collection 2 (after 50% of the neoadjuvant treatment), n: 20; collection 3 (after the end of neoadjuvant treatment and before surgery), n: 14.

**Figure 5 biomolecules-14-01580-f005:**
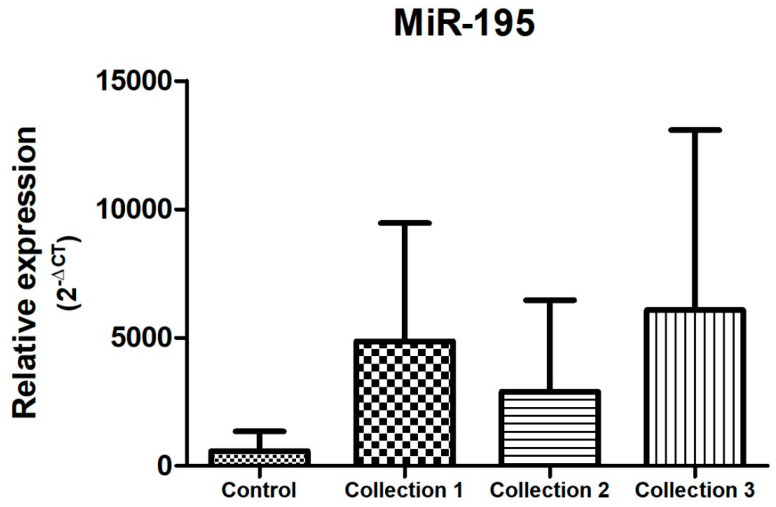
Relative expression of miR-195. Total N: 85 samples; control, n: 29; collection 1 (before neoadjuvant treatment), n: 22; collection 2 (after 50% of the neoadjuvant treatment), n: 20; collection 3 (after the end of neoadjuvant treatment and before surgery), n: 14.

**Figure 6 biomolecules-14-01580-f006:**
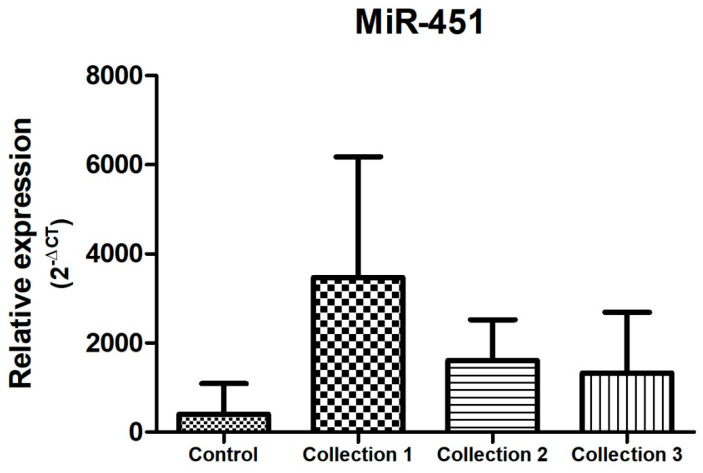
Relative expression of miR-451. Total N: 85 samples; control, n: 29; collection 1 (before neoadjuvant treatment), n: 22; collection 2 (after 50% of the neoadjuvant treatment), n: 20; collection 3 (after the end of neoadjuvant treatment and before surgery), n: 14.

**Figure 7 biomolecules-14-01580-f007:**
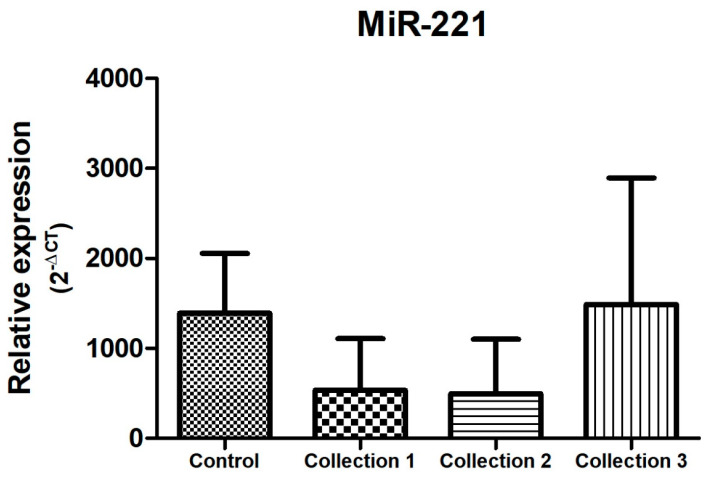
Relative expression of miR-221. Total N: 85 samples; control, n: 29; collection 1 (before neoadjuvant treatment), n: 22; collection 2 (after 50% of the neoadjuvant treatment), n: 20; collection 3 (after the end of neoadjuvant treatment and before surgery), n: 14.

**Table 1 biomolecules-14-01580-t001:** Description of the tumor profile and treatment of the study group participants (n: 27).

Variable	N	% (Approximate)
Age (minimum–maximum/mean/standard deviation)	34–73/50, 11/9, 72	--
Histology		
Non-special subtype carcinoma	26	96.3%
Lobular carcinoma	1	3.7%
Grade of the tumor		
G2	23	85.5%
G3	4	14.5%
Staging		
T:T1//T2//T3//T4	1//2//20//4	3.7%//7.4%//74.1%//14.8%
N:N0//N1//N2//N3	14//11//2//0	51.8%//40.8%//7.4%//0%
M:M0//M1	27//0	100%//0%
Molecular Profiling		
Luminal A	4	14.5%
Luminal B	7	26%
HER-2+/HR ** -	2	7,4%
HER-2+/HR ** +	7	26%
Triple negative	7	26%
Ki67		
<14%	5	18.5%
≤14–50%	18	67%
≥50%	4	14.5%
Chemotherapy		
AC-T ***	11	41.7%
TC ****	4	14.5%
AC-TH *****	4	14.5%
TCH ******	4	14.5%
TC-AC *******	2	7.4%
Others	2	7.4%

** HR: hormone receptor; AC-T ***: doxorubicin + cyclophosphamide—paclitaxel; TC ****: carboplatin + docetaxel; AC-TH *****: doxorubicin + cyclophosphamide—paclitaxel + trastuzumab; TCH ******: carboplatin + docetaxel + trastuzumab; TC-AC *******: carboplatin + paclitaxel—doxorubicin + cyclophosphamide. HER2 -: negative for HER2 proteinHER2 +: positive for HER2 protein

**Table 2 biomolecules-14-01580-t002:** Description of the data regarding the response to neoadjuvant treatment and the surgical treatment performed in the participants of the study group (n: 27).

Variable	N	% (Approximate)
**Clinical Response**		
Complete	12	44,4%
Partial	12	44 %
Stable disease	2	7.4%
Disease progression	1	3.7%
**Pathological Response**		
Complete	6	22.2%
Partial	14	51.8%
No response	2	7.4%
Data not available	5	18.6%

Thirty-three participants without previous oncological diagnosis were included in the control group, all female, aged between 33 and 72 years (mean of 50.5 years, with standard deviation of 9.64).

## Data Availability

All data generated or analyzed during this study are included in this published article.

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
