# Peer review of "Dysregulation of MiR-21, MiR-221 and MiR-451 During Neoadjuvant Treatment of Breast Cancer: A Prospective Study"

_biomolecules, 2024, doi:10.3390/biom14121580_

Round 1
Reviewer 1 Report
Comments and Suggestions for Authors
Here are my recommendations:
1. It is not clear in the manuscript hat is the endogenous control and what is used to normalize the expression of the miR? If its U6 then it also shown change in expression in different groups.
2. Please insert 2-3 more references on page 2 lines 56-59 with “Joint analyses of studies........negative tumors and isolated HER-2 positive”
3. Please insert references on page 2 lines 73-74 with “Due to the greater stability of the microRNA molecule........models as new biomarkers.”
4. Please insert references on page 3 lines 106-108 with “Despite some studies describing the relation between neoadjuvant treatment ............ regimens different from those performed in the Western world.”
5. Please insert references on page 12 lines 369-374.
6. Reference missing on page 13 lines 426-427 “MiR-21 has been evaluated in several studies in breast cancer, ........ compared to control samples from healthy individuals”
7. Please go through the manuscript for the typos and grammatical errors and rectify them.
Reviewer 2 Report
Comments and Suggestions for Authors
The main focus of the paper on the relation between DE circulating miRNAs and BC patients treated with neoadjuvant therapies is interesting. Indeed, there are few scientific papers on the discovery of biological markers linked to neoadjuvant treatment. Despite this evidence, there are some points that need to be clearer: it is not clear which cancer therapies shown in the paper act on which cell cancer pathways if they have the same behavior in terms of cell function. Moreover, there is no table o figure to clearer the biological and functional link between the neoadjuvant treatment and the potential intervention of miRNA in cancer cell. Another point that could help the comprehension of the paper would be the evaluation of potential miRNas interactions of BC patients, also if with the help of predictors of miRNA-gene interactions.
There are some typos errors in the text as instance the sentence "triplo negativo" in table 1 that has to be corrected in the English form "triple negative"
Round 2
Reviewer 2 Report
Comments and Suggestions for Authors
Authors satisfied all my requests and clarify some points described in introduction and discussion sections